# Non-Invasive Assessment of Vascular Circulation Based on Flow Mediated Skin Fluorescence (FMSF)

**DOI:** 10.3390/biology12030385

**Published:** 2023-02-28

**Authors:** Andrzej Marcinek, Joanna Katarzynska, Leslaw Sieron, Robert Skokowski, Jacek Zielinski, Jerzy Gebicki

**Affiliations:** 1Institute of Applied Radiation Chemistry, Lodz University of Technology, 90-924 Lodz, Poland; 2Angionica Ltd., 90-924 Lodz, Poland; 3Department of Athletics, Strength and Conditioning, Poznan University of Physical Education, 61-871 Poznan, Poland

**Keywords:** flow mediated skin fluorescence, FMSF technique, NADH fluorescence, microcirculation, flowmotion, reactive hyperemia response, RHR, hypoxia sensitivity, HS, normoxia oscillatory index, NOI

## Abstract

**Simple Summary:**

The Flow Mediated Skin Fluorescence (FMSF) method, based on the observation of the fluorescence of nicotinamide adenine dinucleotide (NADH) coenzyme from skin tissue, allows for a non-invasive assessment of vascular circulation. The method characterizes dynamic changes in the NADH fluorescence caused by microcirculatory oscillations of endothelial, neurogenic or myogenic origin as well as those observed in response to the post-occlusive reactive hyperemia. It enables assessment of both vasoconstriction and vasodilation. The method defines three major parameters characterizing vascular circulation: reactive hyperemia response, hypoxia sensitivity and normoxia oscillatory index. The FMSF technique seems to be an optimal tool for characterizing macro- and micro-circulation status in a wide range of populations, from healthy physically active people to patients suffering from serious health problems related to vascular dysfunction.

**Abstract:**

Flow Mediated Skin Fluorescence (FMSF) is a new non-invasive method for assessing vascular circulation and/or metabolic regulation. It enables assessment of both vasoconstriction and vasodilation. The method measures stimulation of the circulation in response to post-occlusive reactive hyperemia (PORH). It analyzes the dynamical changes in the emission of NADH fluorescence from skin tissue, providing the information on mitochondrial metabolic status and intracellular oxygen delivery through the circulatory system. Assessment of the vascular state using the FMSF technique is based on three parameters: reactive hyperemia response (RHR), hypoxia sensitivity (HS), and normoxia oscillatory index (NOI). The RHR and HS parameters determine the risk of vascular circulatory disorders and are the main diagnostic parameters. The NOI parameter is an auxiliary parameter for evaluating the state of microcirculation under stress of various origins (e.g., emotional stress, physical exhaustion, or post-infection stress). The clinical data show that the risk of vascular complications is limited among people whose RHR, log(HS), and NOI parameters are not significantly below the mean values determined by the FMSF technique, especially if they simultaneously meet the conditions RHR > 30% and log(HS) > 1.5 (HS > 30), and NOI > 60%.

## 1. Fundamentals of the FMSF Methodology

Lifestyle diseases, including cardiovascular diseases, are the most common cause of death in industrialized countries. Their treatment incurs tremendous cost to the healthcare system. Since current approaches focus mainly on managing symptoms, there is a need for methods and tools for early diagnosis and monitoring of treatment, especially minimally invasive techniques.

There is convincing evidence of the involvement of endothelial dysfunction in the pathogenesis of vascular diseases. Vascular endothelium is an important metabolically active organ that regulates numerous functions and maintains vascular homeostasis, as well as vascular tone and structure—i.e., the dimensions, elasticity, permeability, and reactivity of arteries and veins, including their ability to constrict and dilate. Healthy endothelium is linked to an intact balance of the release of endothelium-derived vasodilating factors (nitric oxide (NO), prostacyclin (PGI2)), endothelium-derived hyperpolarizing factors (EDHFs), and endothelium-derived vasoconstricting factors (endothelin 1, angiotensin II, and thromboxane A2). Under pathological conditions, such as hypercholesterolemia, hypertension, and coronary disease, the production of vasodilating factors is impaired and the production of constricting factors is activated, which also promotes vascular inflammation, remodeling, and thrombosis. A healthy endothelium is essential for undisturbed functioning of the cardiovascular system, while endothelial-dependent dysfunction has been identified as a common thread linking all cardiovascular risk factors. As both an early event and major risk factor, endothelial dysfunction is an important indicator for medical diagnosis. Endothelial dysfunction may be regarded as a barometer of cardiovascular risk [1,2,3,4].

Most studies focus on assessment of endothelial function in conduit arteries. However, there is compelling evidence that functional changes in circulation in other organs and the pathogenesis of cardiovascular diseases are correlated with microcirculation dysfunction, which may precede endothelial impairment in large blood vessels [5]. Microcirculation is that part of the cardiovascular system located between the arterial and venous systems, where intensive gas and metabolic exchanges take place. The response of the microvascular vessels to ischemia is crucial to limit the degree of tissue damage. Numerous studies have shown that the skin microcirculation, which is a readily accessible vascular bed, is representative for the assessment of the systemic microcirculation, its dysfunction, and pathologies [6]. It has been shown that the reactivity of the skin microcirculation is disturbed in persons with an increased risk of ischemic heart disease [7]. The cutaneous microcirculation is also considered to be an independent predictor of atherosclerotic damage [8]. The epidermal layer of the skin is not vascularized, so oxygen and nutrients are transported from the dermis by diffusion. This makes epidermal cells a sensitive marker of early disorders of vascular circulation.

Since direct observation of microcirculatory vessels is difficult, microvascular function in the human skin is mainly studied based on skin blood flow, with the use of such techniques as Laser Doppler Flowmetry (LDF) and Laser Speckle Contrast Imaging (LSCI). However, it is possible to also test microvascular blood circulation via changes in skin biochemistry, especially the mitochondrial NADH redox state of the dermal and/or epidermal cells, which depends on blood circulation and is sensitive to its changes. Based on this effect, we have developed a new, non-invasive diagnostic technique called Flow Mediated Skin Fluorescence (FMSF) for assessment of vascular circulation and metabolic regulation. The FMSF technique is based on measurement of nicotinamide adenine dinucleotide (NADH) fluorescence from skin tissue cells. NADH and its oxidized form (NAD^+^) play a crucial role in biological systems, as redox coenzymes. The interconversion of the NADH/NAD^+^ couple has been studied extensively, including from the mechanistic point of view [9].

As shown in Figure 1, only the reduced NADH form emits significant fluorescence, so the changes in the redox equilibrium of the NADH/NAD^+^ couple can be monitored by measurements of skin fluorescence. The fluorescence of NADH is the strongest component in the overall fluorescence emitted from human skin. It is important to emphasize that the penetration of exciting light (340 nm, maximum of NADH absorption band) in skin tissue is low (about 0.3–0.5 mm). A substantial fraction of the exciting light is therefore absorbed by the epidermis and papillary dermis [10,11,12,13].

NADH fluorescence has been widely used to determine mitochondrial function in vivo [15,16]. The metabolism of keratinocytes in human skin has also been studied in terms of NADH fluorescence and mitochondrial dynamics [17,18].

As circulation at rest rarely provides significant information on its proper functioning or dysfunction, the FMSF technique, like LDF, uses mechanical stimulation to assess vascular status. It measures changes in the intensity of NADH fluorescence from the skin in response to post-occlusive reactive hyperemia (PORH), which is accomplished by blocking and releasing blood flow in the forearm using a typical occlusion cuff (as used for blood pressure measurement devices). This enables assessment of both vasoconstriction and vasodilation.

PORH is the most popular test used to assess vascular reactivity in both macro and microcirculation, based on the use of mechanical provocation. Studies on the mechanism of action of PORH prove that several important mediators are involved, with nitric oxide (NO) acting as a potent vasodilator of muscle-type arteries [19].

One of the most extensively investigated non-invasive methods using PORH is Flow Mediated Dilation (FMD), which is based on monitoring the diameter of brachial arteries after reactive hyperemia with a two-dimensional ultrasound and Doppler ultrasound [20]. This method has been applied widely in clinics and in large-scale epidemiological research, proving that endothelial dysfunction is associated with cardiovascular diseases. It has been shown in many clinical trials that brachial artery FMD independently predicts the risk of future cardiovascular events and enables monitoring of the effectiveness of treatment [21,22,23,24,25,26,27,28]. Low-Flow Mediated Constriction (L-FMC) can be considered as a complementary method and in combination with FMD gives an integrated score of stimulated vascular function [29,30,31].

The PORH test accompanied by observation of blood flow changes in the skin microcirculation (by Doppler methods or tonometry, such as Peripheral Arterial Tonometry (PAT)) is also a popular method of assessing endothelial functioning. The blockage of blood flow in main arteries and its subsequent resumption cause quantitative and dynamic changes related to the bioavailability of NO. Laser Doppler Flowmetry assesses blood flow with a high sampling frequency and very good time resolution, enabling analysis of changes in the skin perfusion after stimulation of the microcirculation [6,32]. The time evolution of the post-occlusive signal resembles that observed by FMD. A major disadvantage of LDF is its poor reproducibility, which is caused mainly by the regional heterogeneity of skin perfusion due to skin anatomy and the small size of the monitored region [33].

NADH fluorescence from human skin at rest provides limited information on the level of the NADH coenzyme in skin cells and the redox equilibrium of the NADH/NAD^+^ pair, as indicators of mitochondrial function. However, the reduced form of the coenzyme (NADH) accumulates under ischemic conditions and during hypoxia, and undergoes oxidation during hyperemia. The degree of change in NADH/NAD^+^ imbalance as a result of the PORH test can be used to assess the vascular response to ischemic conditions. In the last decade, substantial evidence has linked vascular diseases with a dysfunctional response to hypoxia.

Some similarity in laser Doppler and FMSF post-occlusive signals can be observed, although dynamical changes in NADH fluorescence are slower than the dynamic changes of skin blood flow observed by simultaneous FMSF and LDF measurements. In fact, the measured changes in NADH fluorescence reflect the intrinsic changes of NADH levels in skin tissue, providing important information on the mitochondrial metabolic state in terms of oxygen delivery. Both LDF and PAT measurements remain blind to the ischemic period. In contrast, in the FMSF technique the temporal evolution of the ischemic and hyperemic periods resemble those observed by L-FMC and FMD, respectively. Monitoring the changes in NADH fluorescence from the skin reflecting chances of oxygen and nutrient delivery to epidermal cells by the vascular circulation seems an attractive alternative of the circulatory system monitoring without requiring direct observation of the blood flow itself [13,34].

Observations of blood flow changes in the skin microcirculation or NADH fluorescence from epidermal cells by blocking/restoring the blood flow in the brachial artery raises the question of whether macro- and micro-circulation responses to PORH can be distinguished in the overall response to hypoxia.

Blood flow in microvessels is not a homogeneous process, constant over time between successive heart contractions. To be effective, it must be accompanied by vascular oscillations. Oscillations in the microcirculation, known as flowmotion, are a well-recognized feature of cutaneous blood flow [35,36]. The mechanistic aspects of flowmotion have been the subject of extensive study [37]. The main assessment techniques used are Doppler blood flow tests, using for example a Laser Doppler Flowmeter (LDF), which allow for semi-quantitative characterization of changes in cutaneous blood flow [38,39]. Analysis of the signal recorded with an LDF in the frequency domain makes it possible to extract the components of microcirculation oscillations, which are classified as follows: endogenous, independent of NO (<0.0095 Hz); endogenous NO-dependent (0.0095–0.021 Hz); neurogenic (0.021–0.052 Hz); myogenic (0.052–0.15 Hz); respiratory (0.15–0.62 Hz) and cardiac (0.62–2.00 Hz) [40,41].

Vascular changes are determined by changes in the intracellular calcium level and by the membrane potential of vascular smooth muscle cells. The most important processes related to Ca^2+^ activity include the regulation of muscle contraction and, indirectly, the activation of ion pumps, numerous enzymes, and other target proteins including NO synthase, which determines the release of nitric oxide. Nitric oxide synthase, as a result of increased demand for oxygen, also activates the vascular endothelial growth factor (VEGF) receptor. Hypoxia also activates the hypoxia inducible factor (HIF), which is the stimulus that accelerates the transcription of the VEGF gene. Stabilization of HIF-1α is responsible for the regulation of many physiological processes, including wound healing and adaptation to high-altitudes or physical exercise [42].

Myogenic microcirculatory oscillations, associated with changes in the diameter of microvessels (vasomotion), are a very sensitive measure of the microcirculatory response to hypoxia and can be monitored with high precision using the FMSF technique. Thus, the FMSF technique in conjunction with the use of PORH test allows diagnostics of vascular circulation including not only on dysfunctional blood flow in major arteries, but also microcirculatory response to hypoxia.

## 2. Technical Aspects of the FMSF Measurement

The FMSF method measures NADH fluorescence from the skin of the forearm in response to blocking and releasing blood flow. Blood flow is blocked in the forearm using a typical occlusion cuff, commonly used to measure blood pressure. Measurements are performed using the AngioExpert, a device constructed by Angionica Ltd. (Lodz, Poland).

The AngioExpert measures the NADH fluorescence (the emitted wavelength of NADH fluorescence is around 460 nm) at a sampling frequency of 25 Hz, excited by ultraviolet radiation with a wavelength of 340 nm [14]. The penetration of exciting light (340 nm) in skin tissue is low (about 0.3–0.5 mm). A substantial fraction of the exciting light is therefore absorbed by the epidermis and papillary dermis [11]. In these skin regions, the density of blood microvessels is low and the changes in NADH fluorescence depend mainly on the supply of oxygen diffused from deeper layers. The inefficient absorption of the excited light by blood components is limited.

For fluorescence measurements, AngioExpert uses a light-emitting diode (LED-UV) with an emission maximum of approx. 340 nm as the NADH excitation light source and a second photodiode as the NADH fluorescence detector at 460 nm. In the optical head, a set of interference filters is used to ensure proper selection of excitation and emitted light. This is an important aspect for the selective observation of the changes in NADH fluorescence, although alone it would not be sufficient to limit the excitation of other endogenous fluorophores present in the skin, such as collagen, tryptophan, elastin, flavin adenine dinucleotide or some glycated products present. However, to a large extent, such limitation is achieved as a result of NADH excitation primarily in the keratinocytes of the epidermal layer. We have shown in our methodological work [14], that upon illumination of the skin with the 340 nm light the fluorescence of NADH dominates in the emission spectrum. What is more, the changes in fluorescence caused by brachial artery occlusion (increase of approx. up to 15–20% of the total emission at 460 nm) can be selectively attributed to the changes in NADH fluorescence, and it is worth noting, that FMSF method is based mainly on the analysis of the changes in fluorescence at 460 nm.

The use of the photodiode technique enables miniaturization of the system and noise reduction. The power of the LED-UV diode does not exceed 8 mW, so the emitted UV radiation does not pose any danger to the skin, even with prolonged exposure. A scheme of the device is shown in Figure 2 [14].

The test is performed with the patient in a comfortable sitting position, after a minimum adaptation period of 5 min, in a quiet room with a controlled air temperature (24 ± 1 °C). The resting NADH fluorescence value emitted by the epidermal layer of the forearm is recorded for the first 3 min (180 s). The brachial artery is then occluded by inflating the cuff of the device to 60 mm Hg above the systolic pressure. The ischemic response is recorded over a period of 3 min (180 s). During this time, ischemic changes in the NADH fluorescence signal are recorded. Upon completion of occlusion, the cuff pressure is released abruptly, restoring flow in the brachial artery and inducing a hyperemic response for a minimum duration of 4 min (240 s).

Although there is no time limit for fluorescence measurements, the time available for measurement and analysis is constrained by the requirement that the patient remain still during the examination, to avoid artifacts and resulting errors. The data are analyzed using analytical software installed on the AngioExpert device or commercially available programs. The initial section of the measurement, during which the patient adjusts to the examination conditions, is discarded from the analysis. Because cuff occlusion is limited to 3 min and the hyperemic/reperfusion period following occlusion also lasts 3 min, analysis of the signal during ischemic and hyperemic periods is limited to that time.

## 3. Definition of the Measured FMSF Parameters

### 3.1. Reactive Hyperemia Response (RHR)

Figure 3 presents a typical fluorescence signal for a healthy individual, recorded using the FMSF method by the AngioExpert. During the initial stage of the measurements, the baseline is collected (for 3 min). The FMSF signal is normalized with respect to the mean value of the fluorescence in first 1–2 min of this stage of the measurements. Normalization of the signal makes the result of its analysis independent of measurement conditions related to the individual characteristics of the patient’s skin (for example, skin pigmentation, suntan) and/or different technical reasons, as only relative changes are analyzed.

In the second stage of measurements, known as the ischemic response (IR), an increase in NADH fluorescence is observed due to the occlusion of the brachial artery as the cuff is inflated to 60 mm Hg above the systolic blood pressure of the patient. After 3 min, the cuff pressure is released and the NADH fluorescence falls below the baseline, reaching a minimum followed by a return to the baseline. This third measurement stage, called the hyperemic response (HR), consists of a very rapid decrease in NADH fluorescence due to hyperemia (20–30 s) followed by a slow return of NADH fluorescence to baseline due to reperfusion (approximately 3 min).

Based on the combined response from both the ischemic and hyperemic parts of the measured FMSF trace, a Reactive Hyperemia Response (RHR) parameter can be defined [43]. This is a powerful diagnostic tool for characterization of vascular circulation. The RHR parameter characterizes endothelial function related predominantly to the changes in the production of nitric oxide (NO) in the vasculature, mainly in the macrocirculation, due to ischemia and reactive hyperemia (RHR = IR_max_ + HR_max_).

Some additional parameters can be defined that combine the magnitude of changes observed with the rate of NADH fluorescence growth (IR_index_) during ischemia and the return of fluorescence to the baseline after hyperemia (HR_index_). In some cases (some patients with type 1 diabetes), the final level of fluorescence does not reach the level of the baseline. This difference is called the Metabolic Recovery (MR) parameter. A detailed definition of these parameters can be found elsewhere [44].

All parameters mentioned above measure the relative changes in NADH fluorescence, expressed as percentages. Using this approach, perturbations caused by the variability of the skin condition are avoided, and comparisons can be made between unhealthy and healthy populations, or between patients.

### 3.2. Hypoxia Sensitivity (HS)

As can be seen in Figure 3, the FMSF signal oscillates both at rest (basal oscillations called flowmotion at rest (FM)) and even more strongly during the reperfusion period (called flowmotion at the reperfusion period (FM(R)). During the ischemic stage, the signal remains quite smooth because of the blockage of the blood flow, especially in the microcirculation. The altered strength and frequency of oscillations after post-occlusive reactive hyperemia (PORH) reflects the reaction of the vascular microcirculation to hypoxia caused by transient ischemia. Using FMSF signal normalization, two methods can be used to assess the strength of microcirculatory oscillations. The first is based on evaluating the oscillations in terms of the mean square error (MSE), which describes the deviations of the experimental signal points (at a sampling frequency of 25 Hz) from the baseline. In most cases, the baseline around which the FMSF signal oscillates (corresponding to the average fluorescence characteristic for a given patient at rest) is straight. However, in some cases it deviates from a straight horizontal line. Moreover, since during hyperemia the baseline is always an ascending line reaching a plateau the baselines can be defined using the second order polynomial regression method. The mean deviation of the fluorescence signal from the baseline can be used as a measure of the mean magnitude of oscillations. This parameter is objective and patient specific. As MSE values are extremely low, the FM parameters are defined as the MSE values multiplied by a factor of 10^6^, to keep them in the number range of units to hundreds. As mentioned, the fluorescence changes are normalized so the FM parameters remain unitless values. The oscillations of the fluorescence signal (seen in Figure 3) relative to baseline, both for the signal before and after occlusion of the brachial artery, are shown in Figure 4a,c.

The second assessment of the strength of oscillations contained in the FMSF signal can be performed using the fast Fourier transform (FFT) algorithm. Power spectral density (PSD) calculated as a mean squared amplitude with rectangular windowing is very well correlated with flowmotion parameters (FM(R)) defined above (*r* = 0.996). Fast Fourier transform analysis provides an estimate of the signal power at a given frequency and its relative contribution to the total power of the signal. The calculated power is grouped into three frequency intervals: ≤0.021 Hz, (0.021–0.052 Hz) and (0.052–0.15 Hz). These correspond to endothelial, neurogenic, and myogenic activities, respectively (see Figure 4b).

Among low frequency oscillations, the fraction of FM(R) (or PSD × 10^6^) values covering the intensity of flowmotion related to myogenic oscillations (0.052–0.15 Hz) is especially interesting. Recorded during reperfusion (see Figure 4c,d), this value shows what is called Hypoxia Sensitivity (HS), as it is entirely responsible for the increased activity of the vessels after post-occlusive reactive hyperemia. Thus, the HS parameter, similarly to efficient stabilization of HIF-1α in microvascular smooth muscle cells during transient hypoxia, reflects the microcirculatory response to hypoxia.

Whereas the HS parameter varies within quite a broad range, log(HS) remains normally distributed.

### 3.3. Normoxia Oscillatory Index (NOI)

Although the microcirculation at rest rarely provides significant information about its normal functioning or dysfunction, which requires the use of provocations such as PORH, some exceptions seem to be of particular interest. One such exception may be the analysis of the flowmotion at rest, especially the relative ratio of endothelial and neurogenic oscillations to myogenic oscillations. Thus, a new parameter representing the contribution of endothelial and neurogenic oscillations relative to all oscillations detected at low frequency intervals (<0.15 Hz) can be introduced [45]:NOI = [PSD(endothelial) + PSD (neurogenic)]/[PSD(endothelial) + PSD (neurogenic) + PSD(myogenic)] × 100%

Despite of the decrease of flowmotion (FM) at rest with age, the NOI parameter remains age-independent. Moreover, in patients with some diseases affecting the vascular system and thus also basal flowmotion, such as diabetes mellitus (group C–diabetes type-2 patients, n = 70 (40 m, 32 f), mean age 63.1 (45–80 y.)), a similar distribution of NOI is observed as for healthy subjects (group B–healthy middle-aged individuals, n = 32 (19 m, 13 f), mean age 38.2 (30–50 y.)) (see Figure 5) [46].

As has been shown for hundreds of patients investigated using the FMSF method, less than 15% of individuals have an NOI parameter below 60%.

As will be discussed in more detail, there is convincing evidence that a chronic decrease in NOI is associated with various types of stress, such as emotional stress, physical exhaustion, or post-infection stress. It is important to mention that such deviation from the normal NOI distribution may be the result of a significant decrease in endothelial and neurogenic oscillations, with a relatively unchanged value of myogenic oscillations, or, conversely, a significant increase in the myogenic component of basal oscillations at rest.

RHR and log(HS) are the key diagnostic parameters derived from FMSF measurements. They can be used for efficient characterization of vascular circulation based on the response to transient ischemia. NOI is an auxiliary parameter to assess the state of microcirculation under stress of various origins. Chronically low NOI values can lead to the development of serious vascular circulatory disorders.

### 3.4. Cardiac Oscillations

Heartbeat oscillations (0.6–2 Hz) are also seen in the FMSF signal and PSD, but they remain weak compared to the total PSD and will not be discussed further. It is worth noting, however, that what may appear to be weak signal noise in Figure 4a,c is largely related to cardiac oscillations (see Figure 4e,f).

The weak effect of the heartbeat on changes in total cutaneous NADH fluorescence confirms that a large fraction of the exciting light (340 nm) does not reach the blood vessels directly, as it is mainly absorbed by the epidermis [17]. Skin model Monte Carlo simulations have confirmed that NADH fluorescence is only very weakly affected by blood volume in skin tissue [13].

## 4. Diagnostic Potential of the FMSF Method

By interpreting the parameters and the dynamics of the NADH fluorescence signal emitted from skin cells in response to occlusion ischemia and subsequent hyperemia, it is possible to identify vascular disorders that can lead to the development of chronic diseases.

### 4.1. Reactive Hyperemia Response (RHR)

Based on the reactive hyperemia response (RHR parameter) to transient ischemia, the condition of vascular circulation found for three different groups of patients can be distinguished with high statistical significance: A–endurance athletes, n = 50 (33 m, 17 f), mean age 22.0 (16–35 y.); B–healthy middle-aged individuals, n = 32 (19 m, 13 f), mean age 38.2 (30–50 y.); C–diabetes type 2 patients, n = 70 (40 m, 32 f), mean age 63.1 (45–80 y.) (Figure 6). The results presented in Figure 6 b show that individuals with low and high RHR values can be found in each group, but the proportion between the individual RHR ranges is clearly different. In group A, patients with RHR below 25% are the exception, whereas in group C they constitute the dominant majority, while patients with RHR ≥ 35% are the exception [43].

Diabetes mellitus (DM) is a chronic metabolic disease that causes vascular complications over time. Excessive production of mitochondrial reactive oxygen species (ROS) is a central mechanism for the development of diabetes complications. It is hypothesized that ROS overproduction may be secondary to impaired responses to hypoxia, due to the inhibition of the hypoxia-inducible factor HIF-1α by hyperglycemia [47,48]. One of the most frequent and severe microvascular complications of DM is diabetic kidney disease (DKD). It has been suggested that vascular endothelial growth factor (VEGF) is critical in glomerular physiology, while DKD is characterized by an imbalance in the expression of VEGF and angiopoietin [49]. HIF-1α is directly responsible for up-regulation of VEGF and is closely related to the etiology of DKD [50,51]. The proper response to hypoxia is one of the mechanisms of kidney adaptation to oxygen deficiency [52].

In research at the Medical University of Lodz (Poland), studies were conducted on a group of 84 volunteers, including 30 patients with diabetic kidney disease (DKD), 33 patients with diabetes mellitus without complications, and 21 healthy control subjects. The results showed that all three groups were clearly differentiated by the RHR parameter with a high statistical significance (RHR(control) = (34.37 ± 8.18)% vs RHR(DM w/o complications) = (28.75 ± 7.12)%, *p* < 0.05; RHR(control) = (34.37 ± 8.18)% vs RHR(DKD) = (18.31 ± 5.06)%, *p* < 0.001 [53]. The whole studied group showed a significant correlation between the RHR parameter and nephrological parameters, such as serum creatinine concentration (sCr, *r* = −0.3, *p* < 0.05) and estimated glomerular filtration rate (eGFR, *r* = −0.61, *p* < 0.001). Because of their impaired adaptation to hypoxia, patients with diabetes have a higher risk of CVD, and the risk of kidney damage increases by up to 12–17 times [44,54].

### 4.2. Hypoxia Sensitivity (HS)

A similar conclusion to that formulated on the basis of RHR can be drawn from the analysis of log(HS). The results summarized in Figure 7 were collected for the same three study groups (see Figure 6 for comparison): A–endurance athletes; B–healthy middle-aged individuals; C–diabetes type 2 patients. Again, individuals with low and high log(HS) values can be found in each group, but the proportion between the individual log(HS) ranges is clearly different. Moreover, in group A (highly trained athletes) over 30% of the whole group had HS values above 200 (log(HS) > 2.3). Their microvascular circulation can be described as excellent, and such individuals have strong potential to be outstanding athletes. Similarly, high HS values were hardly ever observed in the other groups. Individuals with very low HS values showed a very weak response to ischemia, which may indicate some microvascular complications. Microvascular complications are unlikely to develop in patients with diabetes type 2 with HS > 100 (log(HS) > 2, less than 10% of the group C population), despite long disease duration and advanced age [55].

The differences between the groups were partly due to the difference in age, as flowmotion was found to decrease with age [56]. However, this does not affect our general conclusions, especially in terms of individuals with very low log(HS) value finding indicative of microvascular complications. It has also been found that FM(R) (and thus log(HS)) is sensitive to both systolic and diastolic blood pressure, as observed for healthy individuals with normalized blood pressure [56]. This is in agreement with the earlier observation that post-ischemic blood flow oscillations are a sensitive measure of microcirculatory dysfunction in humans with essential arterial hypertension [36].

Similarities in the distribution of log(HS) and RHR parameters among different groups of patients seem to suggest a correlation (for the healthy population *r* = 0.413, *p* = 0.0001) [43]. However, for patients with diseases/disorders of vascular origin, this correlation may be false. For example, the changes in RHR may be greater in the case of CVD, where dysfunction in macrocirculation prevails, regardless of log(HS) values, while the log(HS) parameter can effectively be used to determine the chance of healing in patients with diabetic foot ulcers, regardless of the measured value of the RHR parameter.

In the group of patients with diabetic kidney disease, there is a statistically significant decrease in the log(HS) value found for DKD patients in comparison to healthy subjects (log(HS) = 1.03 vs. log(HS) = 1.59, *p* < 0.001). A significant negative correlation between the log(HS) parameter and sCr and a positive correlation with eGFR were found for the whole studied group (sCr, *r* = −0.33, *p* < 0.01; eGFR, *r* = 0.55, *p* < 0.001) [53].

Diabetic foot ulcers (DFU) are among the most common complications affecting patients with diabetes mellitus. It has been suggested that disturbed hypoxic sensitivity may be responsible for impaired wound healing [47,50,57,58,59,60]. Stabilization of HIF-1α may be critical for wound healing, since cellular adaptive responses to hypoxia are mediated by the hypoxia-inducible factor. HIF-1α is also responsible for up-regulation of the expression of vascular endothelial growth factor (VEGF), which is an essential mediator of neovascularization [50]. Since a low log(HS) parameter indicates impaired flowmotion response to hypoxia, it could be also used to predict vascular complications in diabetic foot ulceration.

In a study conducted at the Medical University of Lodz (Poland), 42 patients with DFU and long-term diabetes (over 10 years) were differentiated based on the HS parameter using FMSF. The group of patients with the highest HS values (n = 11, 8m, 3f) did not show any significant disturbances in blood circulation (only one patient had a history of amputation). The second group, composed of patients with the lowest HS values (n = 11, 6m, 5f), showed constant disturbances of vascular circulation. These patients had higher incidences of hypertension, hyperlipidemia, and cardiovascular disease (CVD), as well as neuropathy and nephropathy. Amputations were also more frequent in this group. The HS parameters indicating serious microcirculatory disorders were found to correlate with USG Doppler results indicating macrovascular disturbances (*r* = −0.547, *p* = 0.035) [61]. Since changes in microcirculation occur sooner than in large blood vessels, the log(HS) parameter allows for the early detection of vascular disorders. The FMSF technique can be recommended both for predicting vascular complications in diabetic patients, including DFU, and monitoring their health status.

### 4.3. Normoxia Oscillatory Index (NOI)

In extensive research conducted at Poznan University of Physical Education (Poland), the FMSF method was used on large groups of professional and amateur athletes. In many cases, hypoxia caused strong activation of myogenic oscillations (see Figure 4), indicating an outstanding predisposition for competitive sports. Moreover, a significant reduction of endothelial and neurogenic oscillations was noticed after exercise to exhaustion (an extreme type of test to assess the athlete’s performance) (see Figure 8). High-intensity exercise, accompanied by a corresponding increase in reactive oxygen species (ROS), diminishing NO bioavailability, is known to impose transient stress on the vascular endothelium, as measured by flow-mediated dilatation (FMD) [62,63]. This effect is often called the “exercise paradox.” Strenuous physical exercise also affects calcium homeostasis, and this effect persists until the late recovery phase [64,65].

Figure 8 presents the changes in the fluorescence signal relative to the normalized baseline before and after high intensity exercise and the corresponding Power Spectral Density for a highly trained athlete. An evident decrease in the intensity of flowmotion, especially endothelial (<0.021 Hz) and neurogenic (0.021–0.052 Hz) oscillations, can be seen. The flowmotion normalized after 1–3 h of rest and returned to the values observed before exercise. This observation is in accordance with other studies [62,63].

In the aforementioned study on professional and amateur athletes, the strength of endothelial and neurogenic oscillations was measured on the basis of absolute PSD values before and after exercise. However, the flowmotion of a single patient is not always known before factors occur disturbing microcirculatory oscillations. As flowmotion has been found to decrease with age, the reference ranges in different age groups have not yet been established [56]. However, analysis of oscillations based on the NOI parameter (which is independent of age) can also provide reference values. Figure 9 presents the changes in NOI and RHR parameters following strenuous exercise in the group of 26 highly trained amateur runners (n = 26, 26 m, mean age 27.7 (21–40 y.)), who were participating in various competitive sports (long-distance running, cross-country, marathons) [45]. Physical exhaustion can be seen to have a strong effect on both parameters. It should be emphasized that the NOI parameter and the RHR parameter represent distinctive properties of the vascular system. The NOI parameter characterizes microcirculation as it is related to microcirculatory oscillations. In contrast, the RHR parameter characterizes circulation predominantly in large and medium size arteries, as it reflects NO bioavailability caused by reactive hyperemia. It is evident that both parts of the vascular system are affected by strenuous exercise.

Recent years have been dominated by the global health crisis caused by the COVID-19 pandemic and its consequences. The effects of SARS-CoV-2 infection are not limited to the respiratory tract but include serious injuries to vascular circulation, caused by damage to the vascular endothelium [66,67,68]. Patients recovering from SARS-CoV-2 infection complain of persistent symptoms, the most common of which is fatigue (over 70% of post-COVID patients). There is limited knowledge about the pathophysiology of post-COVID chronic fatigue. Some analogies between post-COVID fatigue syndrome and myalgic encephalomyelitis/chronic fatigue syndrome (ME/CFS) link it to calcium homeostasis dysfunction [69,70,71,72]. Since post-COVID syndrome can be linked to permanent vascular dysfunction (as a consequence of the acute phase of COVID-19 infection), it seems rational to compare the chronic vascular effects of post-COVID fatigue with the transient vascular effects of fatigue caused by high-intensity exercise.

A group of 45 patients (n = 45, 19 m, 26 f, mean age 41.5 (30–50 y.)) with post-COVID syndrome, reporting limited tolerance to exercise and above 50% greater fatigue compared to their pre-COVID levels, was tested using the FMSF technique and compared to a healthy control group (n = 32, 19 m, 13 f, mean age 37.8, (30–50 y.)) [45].

Figure 10 shows a comparison and distribution of the NOI and RHR parameters in the control and post-COVID groups. The vascular effects of post-COVID-related fatigue and exercise-related fatigue measured by the NOI and RHR parameters are quite similar (see Figure 9 and Figure 10). It seems likely that the observed link between the pathophysiology of transient fatigue caused by strenuous exercise and by chronic post-COVID fatigue originates from the modification calcium homeostasis leading to endothelial dysfunction [64,65,69,70].

The analysis of microcirculation oscillations at rest using the FMSF technique can be of particular use for monitoring physical stress related to strenuous exercise or post-infection fatigue. It is likely that a similar methodology can be applied to the vascular consequences of psychological stress. In numerous individual cases, strong myogenic oscillations in flowmotion at rest have been observed, presumably due to the ischemic response caused by microcirculation vasoconstriction accompanying psychological stress associated with, e.g., traumatic events or cancer disease [46]. In these cases, a strong decrease in the NOI value is also observed, although the changes in the distribution of flowmotion oscillations (activation of the myogenic component) are different. This very attractive potential use of FMSF technique, to assess the vascular consequences of stress of different origins, will be explored in future work.

## 5. Concluding Remarks

Flow Mediated Skin Fluorescence (FMSF) is a new non-invasive method enabling assessment of vascular circulation and/or metabolic regulation. The method measures the dynamical changes of NADH fluorescence from the skin tissue, providing information on mitochondrial metabolic status and intracellular oxygen delivery through the circulatory system. It measures stimulation of the circulation in response to post-occlusive reactive hyperemia. It enables assessment of both vasoconstriction and vasodilation.

The FMSF technique is an optimal tool for characterizing macro- and micro-circulation status in a wide range of populations, from healthy physically active people to patients suffering from serious health problems related to vascular dysfunction. Based on three parameters (RHR, log(HS), NOI) measuring the circulatory response to hypoxia, the FMSF technique provides a powerful diagnostic tool for the characterization of vascular circulation.

The diagnostic potential of the RHR parameter includes:−assessing dysfunction of the vascular circulation, especially the macrocirculation;−predicting the risk of developing cardiovascular diseases often comorbid with diabetes mellitus.

The diagnostic potential of the log(HS) parameter includes:


−assessing microcirculatory dysfunction in diabetes, cardiovascular disease, peripheral arterial disease, hypertension;−predicting healing in difficult-to-heal wounds (including diabetic foot ulcers);−assessing microcirculatory status to determine exercise tolerance in healthy and non-healthy people.


The diagnostic potential of the NOI parameter includes:


−assessing fatigue due to stress from various sources,−monitoring microcirculation in the recovery and rehabilitation process.


The clinical data show that the risk of vascular complications is limited among people whose RHR, log(HS), and NOI parameters are not significantly below the average values determined by the FMSF technique, especially if they simultaneously meet the condition RHR > 30% and log(HS) > 1.5 (HS > 30), and NOI > 60%.

## Figures and Tables

**Figure 1 biology-12-00385-f001:**
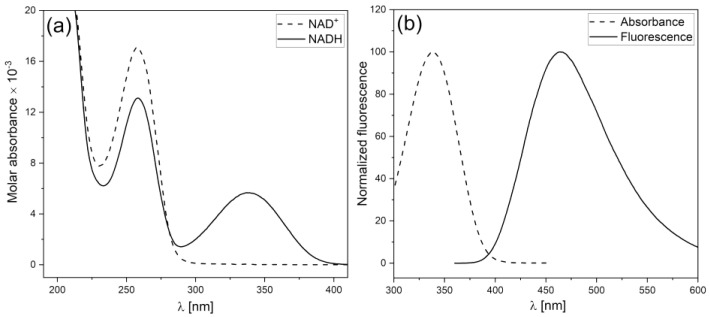
Electronic absorption spectra of NADH and NAD^+^ (**a**) and fluorescence spectra of NADH (**b**) [14]. Reproduced from Katarzynska, J; Lipinski, Z.; Cholewinski, T., Piotrowski, L; Dworzynski, W.; Urbaniak, M.; Borkowska, A.; Cypryk, K.; Purgal, R.; Marcinek, A.; Gebicki J. Non-invasive evaluation of microcirculation and metabolic regulation using flow mediated skin fluorescence (FMSF): Technical aspects and methodology. *Rev Sci Instrum.* **2019**, *90*, 104104; DOI:10.1063/1.5092218, with the permission of AIP Publishing.

**Figure 2 biology-12-00385-f002:**
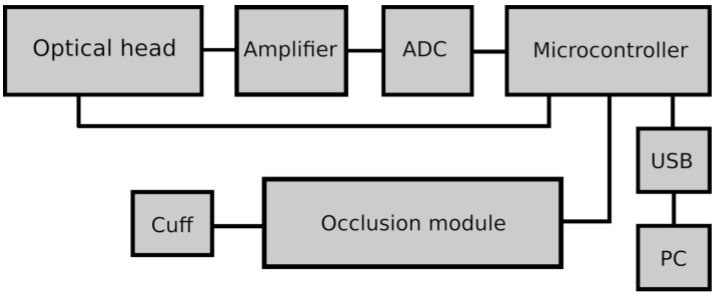
Block scheme of the AngioExpert utilizing FMSF [14]. Reproduced from Katarzynska, J; Lipinski, Z.; Cholewinski, T., Piotrowski, L; Dworzynski, W.; Urbaniak, M.; Borkowska, A.; Cypryk, K.; Purgal, R.; Marcinek, A.; Gebicki J. Non-invasive evaluation of microcirculation and metabolic regulation using flow mediated skin fluorescence (FMSF): Technical aspects and methodology. *Rev Sci. Instrum.* **2019**, *90*, 104104; DOI:10.1063/1.5092218, with the permission of AIP Publishing.

**Figure 3 biology-12-00385-f003:**
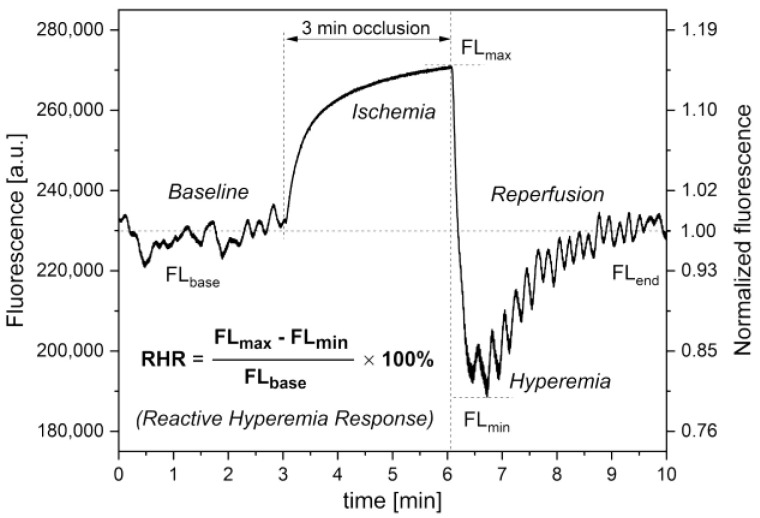
Exemplary FMSF trace recorded for an athlete (male, 40 years); definition of the RHR parameter [43]. Reproduced from: Katarzynska, J.; Zielinski, J.; Marcinek, A.; Gebicki, J. New Approach to Non-Invasive Assessment of Vascular Circulation Based on the Response to Transient Ischemia. *Vasc. Health Risk Manag.* **2022**, *18*, 113–116; DOI:10.2147/VHRM.S358983, Dove Medical Press Ltd. Publisher.

**Figure 4 biology-12-00385-f004:**
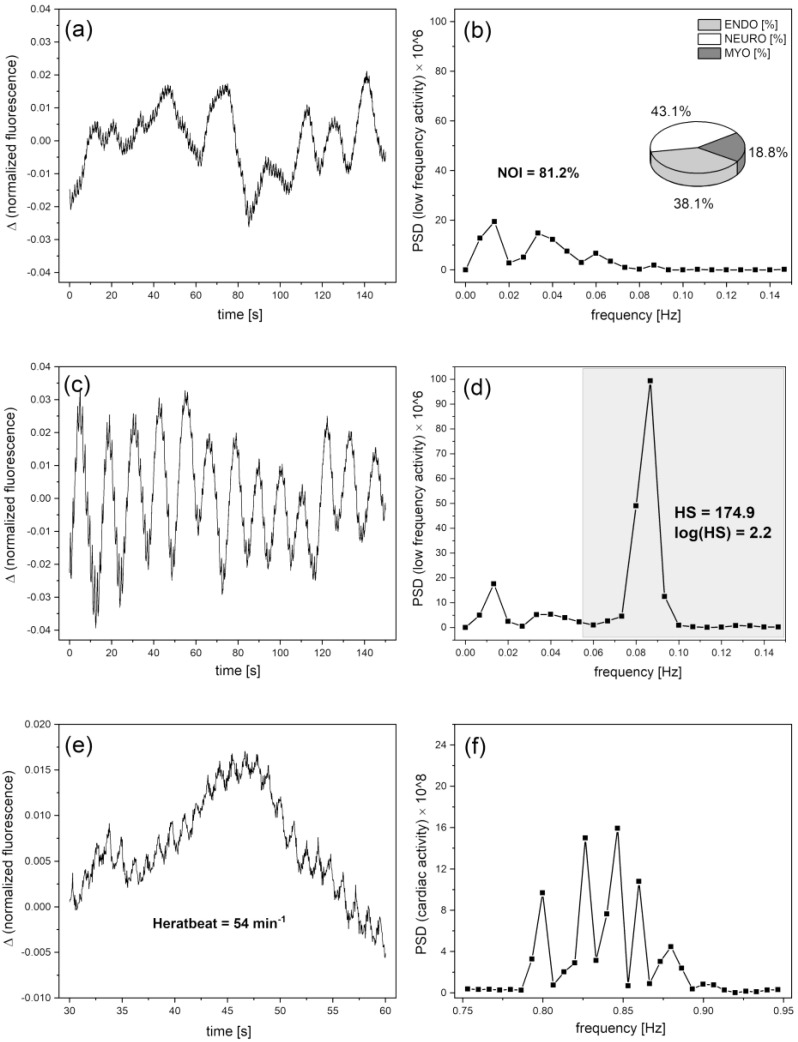
Exemplary oscillations of the FMSF signal relative to baseline with corresponding power spectrum density (PSD) observed for an athlete (male, age 40 years); the pie chart (inset (**b**)) shows the presence of the three components of flowmotion-endothelial (ENDO), neurogenic (NEURO), and myogenic (MYO), respectively: (**a**,**b**) flowmotion at rest; (**b**) definition of NOI [%] = ENDO [%] + NEURO [%] parameter; (**c**,**d**) flowmotion after occlusion during hyperemia and reperfusion; (**d**) definition of HS = PSD(MYO) × 10^6^ parameter; (**e**) heartbeat oscillations seen in the FMSF signal over a 30 s time period; (**f**) cardiac component of the PSD.

**Figure 5 biology-12-00385-f005:**
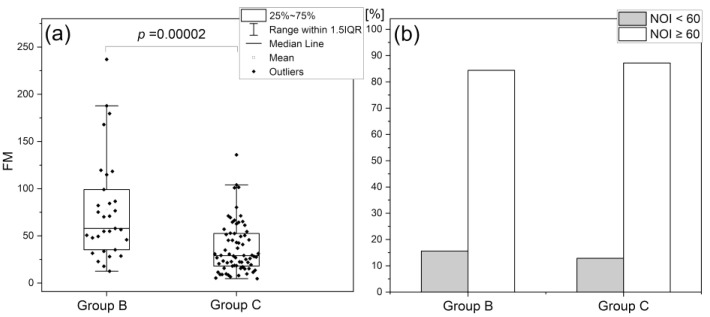
Comparison of the FM (**a**) and NOI (**b**) parameters in groups B–healthy middle-aged individuals, and C–diabetes type-2 patients; the *p*-value was calculated based on the Mann–Whitney test [46]. Based on: Gebicki J, Katarzynska J, Marcinek A. Effect of psychological stress on microcirculation oscillations: diagnostic aspects. *Vasc Health Risk Manag.* **2023**, *19*, 79–82; DOI:10.2147/VHRM.S399082, Dove Medical Press Ltd. publisher.

**Figure 6 biology-12-00385-f006:**
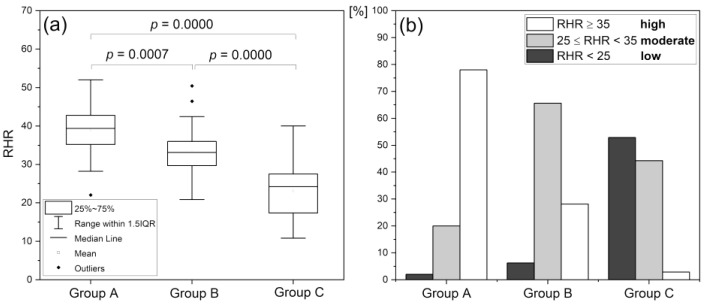
Assessment of RHR parameter in groups A, B, and C: A–highly trained endurance athletes; B–healthy middle-aged individuals; C–diabetes type 2 patients; (**a**) statistical differentiation (one-way ANOVA with the Scheffe post hoc test) of groups A, B, and C based on the RHR parameter; (**b**) distribution of the RHR parameter [43]. Reproduced from: Katarzynska, J.; Zielinski, J.; Marcinek, A.; Gebicki, J. New Approach to Non-Invasive Assessment of Vascular Circulation Based on the Response to Transient Ischemia. *Vasc. Health Risk Manag.* **2022**, *18*, 113–116; DOI:10.2147/VHRM.S358983, Dove Medical Press Ltd. publisher.

**Figure 7 biology-12-00385-f007:**
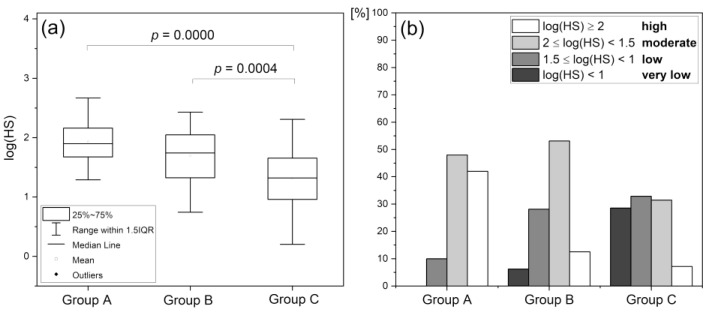
Assessment of log(HS) parameter in various groups: A–highly trained endurance athletes; B–healthy middle-aged individuals; C–diabetes type 2 patients; (**a**) statistical differentiation (one-way ANOVA with the Scheffe post hoc test) of groups A, B, and C based on the log(HS) parameter; (**b**) distribution of the log(HS) parameter [43]. Based on: Katarzynska, J.; Zielinski, J.; Marcinek, A.; Gebicki, J. New Approach to Non-Invasive Assessment of Vascular Circulation Based on the Response to Transient Ischemia. *Vasc. Health Risk Manag.* **2022**, *18*, 113–116; DOI:10.2147/VHRM.S358983, Dove Medical Press Ltd. publisher.

**Figure 8 biology-12-00385-f008:**
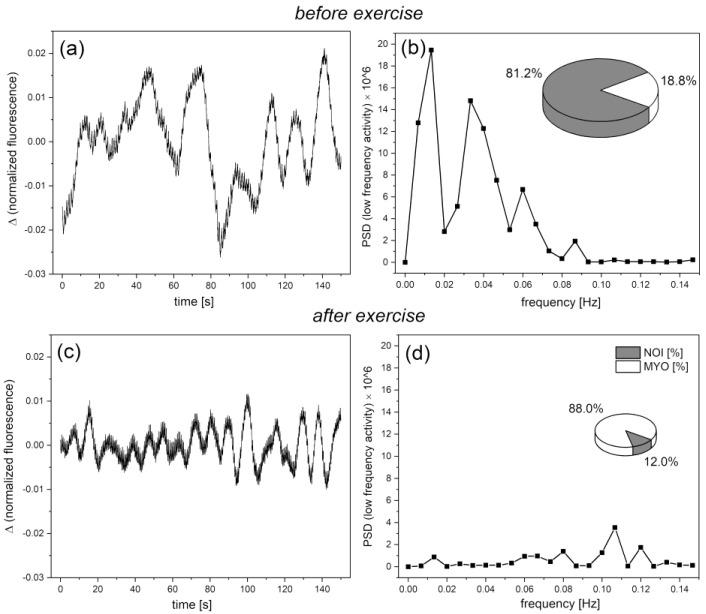
Changes in the oscillations of the FMSF signal recorded for an athlete (male, 40 years) before (**a**) and after (**c**) high-intensity exercise (left) and the corresponding (**b** and **d**) Power Spectral Densities (PSD); the pie charts show the decrease in the NOI parameter [45]. Reproduced from: Chudzik, M.; Cender, A.; Mordaka, R.; Zielinski, J.; Katarzynska, J.; Marcinek, A.; Gebicki, J. Chronic fatigue associated with post-COVID syndrome versus transient fatigue caused by high-intensity exercise: are they comparable in terms of vascular effects? *Vasc Health Risk Manag.* **2022**, *18*, 711–719; DOI:10.2147/VHRM.S371468, Dove Medical Press Ltd. publisher.

**Figure 9 biology-12-00385-f009:**
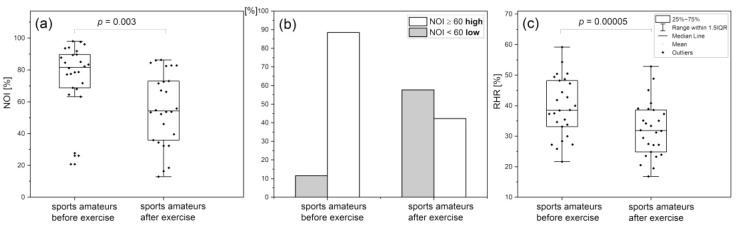
Comparison of NOI (**a**,**b**) and RHR (**c**) parameters in a group of amateur runners before and after strenuous exercise; the *p*-values were calculated by the Wilcoxon signed ranks test (**a**) or the paired sample *t*-test (**c**) [45]. Reproduced from: Chudzik, M.; Cender, A.; Mordaka, R.; Zielinski, J.; Katarzynska, J.; Marcinek, A.; Gebicki, J. Chronic fatigue as-sociated with post-COVID syndrome versus transient fatigue caused by high-intensity exercise: are they com-parable in terms of vascular effects? *Vasc Health Risk Manag.* **2022**, *18*, 711–719; DOI:10.2147/VHRM.S371468, Dove Medical Press Ltd. publisher.

**Figure 10 biology-12-00385-f010:**
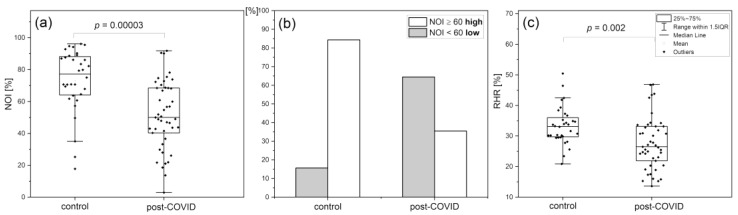
Comparison of NOI (**a**,**b**) and RHR (**c**) parameters in the post-COVID and control group; the *p*-values were calculated by the Mann–Whitney test (**a**) and the two sample *t*-test (**c**) [45]. Reproduced from: Chudzik, M.; Cender, A.; Mordaka, R.; Zielinski, J.; Katarzynska, J.; Marcinek, A.; Gebicki, J. Chronic fatigue as-sociated with post-COVID syndrome versus transient fatigue caused by high-intensity exer-cise: are they com-parable in terms of vascular effects? *Vasc Health Risk Manag.* **2022**, *18*, 711–719; DOI:10.2147/VHRM.S371468, with the permission of Dove Medical Press Ltd.).

## Data Availability

Not applicable.

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
