# Peer review of "Non-Invasive Assessment of Vascular Circulation Based on Flow Mediated Skin Fluorescence (FMSF)"

_biology, 2023, doi:10.3390/biology12030385_

Round 1

Reviewer 1 Report

This is an interesting review discussing the current knowledge on Flow Mediated Skin Fluorescence in the assessment of microcirculation. The paper is clearly written and well prepared. In my opinion, such overview clearly deserve publication as the diagnosis of microvascular dysfunction still remains a serious problem in cardiovascular medicine. In my opinion, the manuscript does not require the serious revision.

Author Response

Thank you very much for your positive comments.

Reviewer 2 Report

In this manuscript, the authors successfully developed a new, non-invasive diagnostic technique called Flow Mediated Skin Fluorescence (FMSF) for assessment of vascular circulation and metabolic regulation. Assessment of the vascular state using the FMSF technique is based on three parameters: RHR, HS, and NOI, Which is an optimal tool for characterizing macro and microcirculation status in a wide range of populations. But more details should be discussed.

1. Figure 3: If the patient is continuously monitored for 10 minutes, will it cause skin damage and photobleaching, resulting in inaccurate detection of fluorescence intensity?

2. Figure 4 (a) shows that the fluorescence intensity drops sharply between 70-80 s. What is the reason? Why the fluorescence intensity starts to rise after 80 s? The author needs to explain in the text.

3. Figure 4 (c and d) shows changes of fluorescence intensity after occlusion during hyperemia and reperfusion, the figures shows that the fluorescence intensity between 80 and 120s has been decreasing slowly, and Figure 4 (a) seems to be opposite after 80s. The change in (a) and (c) should be explained. The same concern is about Figure 8 (a) and (c).

4. As Figure 4 (d) shows, myogenic is only sensitive to hypoxia after reperfusion. What is the difference between endogenous and neural tissue? Need to explain the biological mechanism.

5. The three groups of experiments of RHF and HS shown in the paper are healthy and sick, but the ages of the three groups are completely different. The authors need to confirm whether there is age influence.

Author Response

  1. Photobleaching is not observed and is not expected for monitoring the NADH fluorescence which is main target of the method (see third paragraph in Technical aspects of the FMSF measurements). As can be seen in Figure 3, the NADH fluorescence signal returns to its initial values after occlusion. Only in some cases (some patients with type 1 diabetes), the final level of fluorescence does not reach the level of the baseline (see lines 288-291).

2 and 3. Figure 4 shows only those fragments of the FMSF signal, presented in Figure 3, that contain oscillations. We added the following explanation in line 313: “The oscillations of the fluorescence signal (seen in Fig. 3) relative to such a baseline, both for the signal before and after occlusion of the brachial artery, are shown in Fig. 4a and c.”. We also made some changes in Caption to Figures 4 and 8: “Exemplary oscillations of the FMSF signal relative to baseline …”. The changes in the signal are due to the imposition of the vascular vibrations of different frequencies and of different amplitude (endothelial, neurogenic and myogenic oscillations).

4 and 5. The topic of activation of myogenic oscillations after PORH as well as the dependence of HS on age and blood pressure is addressed in the paper (see lines 179-194, 337-345, 470-477). 

Reviewer 3 Report

NADH fluorescence-based research method is presented in the manuscript. The experiment was done on a large number of patients, although the total number of participants is not presented in the manuscript, but is presented a number of patients in specific cases. Additionally, research has been done on post-COVID patients. The result shows promising potential for utilizing the Flow Mediated Skin Fluorescence method.

I have some comments after reading the manuscript.

1. In the fluorescences method authors utilize UV irradiation at 340 nm with a power 8 mW. Could the authors indicate the officially permitted irradiation power in the manuscript, for comparison?

2. Was ethics permission required for this study?

3. The abstract gives rise to the view that fluorescence is the foundation of all studies. During the reading of the article, this matter does not clearly understandable.

3.1. Could the authors organize and number the formulas by which the calculations were performed, and link to them in the text? (The formulas are in the text now; I mean organize them as formulas)

3.2. The RHR parameter is clearly presented in figure 3. The HS and NOI parameters are not clearly understood. They are based on fluorescence?, or they are other measurements method? Please emphasize in the abstract or manuscript if these parameters are based on NADH fluorescence, or these parameters are calculated by other methods.

3.3. Could the authors specify (illustrate) on the fluorescence graphs in Figure 4 what information should be paid attention to or what data if any, should be taken from the fluorescence graph? For comparison, figure 3 is very clearly illustrated.

Author Response

  1. The work is focused on the FMSF method but not on the technical details of the AngioExpert device.But we can add that the device has obtained all required certificates allowing it to be used and sell as medical device in the EU.
  2. This is a review article referring to the clinical trial results described in details in the original articles. Relevant ethics committee approvals are detailed there.
  3. In fact, the FMSF method described in the article is based solely on the analysis of the NADH fluorescence signal. And we repeat this many times throughout the article.

3.1 and 3.2 All 3 parameters are separately defined in dedicated chapters and Figures: RHR - see chapter Reactive Hyperemia Response (RHR) - lines 255-295, and Figure 3; HS - see Hypoxia Sensitivity(HS) - lines 298-345 and Figure 4; NOI - see Normoxia Oscillatory Index (NOI) and Figure 4 and we do not see the need for their additional numbering.

3.3 Figure 4 shows only those fragments of the FMSF signal, presented in Figure 3, that contain oscillations. We added the following explanation in line 313: “The oscillations of the fluorescence signal (seen in Fig. 3) relative to such a baseline, both for the signal before and after occlusion of the brachial artery, are shown in Fig. 4a and c.”. We also made some changes in Caption to Figures 4 and 8: “Exemplary oscillations of the FMSF signal relative to baseline …”.